# *WT1* Expression Levels Combined with Flow Cytometry Blast Counts for Risk Stratification of Acute Myeloid Leukemia and Myelodysplastic Syndromes

**DOI:** 10.3390/biomedicines9040387

**Published:** 2021-04-06

**Authors:** Valentina Giudice, Marisa Gorrese, Rosa Vitolo, Angela Bertolini, Rossella Marcucci, Bianca Serio, Roberto Guariglia, Idalucia Ferrara, Rita Pepe, Francesca D’Alto, Barbara Izzo, Antonio Pedicini, Nunzia Montuori, Maddalena Langella, Carmine Selleri

**Affiliations:** 1Department of Medicine, Surgery and Dentistry, “Scuola Medica Salernitana”, University of Salerno, 84081 Baronissi, Salerno, Italy; angy.11@hotmail.it (A.B.); cselleri@unisa.it (C.S.); 2Hematology and Transplant Center, University “Hospital San Giovanni di Dio e Ruggi D’Aragona”, 84131 Salerno, Italy; marisa.gorrese@sangiovannieruggi.it (M.G.); vitolo.ross@gmail.com (R.V.); rossellamarcucci89@gmail.com (R.M.); bianca.serio@sangiovannieruggi.it (B.S.); roberto.guariglia@sangiovannieruggi.it (R.G.); idalucia.ferrara@sangiovannieruggi.it (I.F.); rita.pepe@ymail.com (R.P.); daltofrancesca@libero.it (F.D.); maddalena.langella@sangiovannieruggi.it (M.L.); 3Clinical Pharmacology, University Hospital “San Giovanni di Dio e Ruggi D’Aragona”, 84131 Salerno, Italy; 4Department of Molecular Medicine and Medical Biotechnology, CEINGE-Biotecnologie Avanzate, University of Naples “Federico II”, 80138 Naples, Italy; barbara.izzo@unina.it; 5Hematology and Hematopoietic Stem Cell Transplantation Unit, AORN San Giuseppe Moscati, 83100 Avellino, Italy; pemariconda2601@aosgmoscati.av.it; 6Department of Translational Medical Sciences, “Federico II” University, 80138 Naples, Italy; nmontuor@unina.it

**Keywords:** Wilms’ tumor 1, acute myeloid leukemia, myelodysplastic syndrome, prognosis, flow cytometry

## Abstract

Wilm’s tumor 1 (*WT1*), a zinc-finger transcription factor and an epigenetic modifier, is frequently overexpressed in several hematologic disorders and solid tumors, and it has been proposed as diagnostic and prognostic marker of acute myeloid leukemia (AML) and myelodysplastic syndrome (MDS). However, the exact role of *WT1* in leukemogenesis and disease progression remains unclear. In this real-world evidence retrospective study, we investigated prognostic role of WT1-mRNA expression levels in AML and MDS patients and correlations with complete blood counts, flow cytometry counts, and molecular features. A total of 71 patients (AML, *n* = 46; and MDS, *n* = 25) were included in this study, and *WT1* levels were assessed at diagnosis, during treatment and follow-up. We showed that *WT1* expression levels were inversely correlated with normal hemopoiesis in both AML and MDS, and positively associated with blast counts. Flow cytometry was more sensitive and specific in distinguishing normal myeloid cells from neoplastic counterpart even just using linear parameters and CD45 expression. Moreover, we showed that a simple integrated approach combining blast counts by flow cytometry, *FLT3* mutational status, and *WT1* expression levels might be a useful tool for a better prognostic definition in both AML and MDS patients.

## 1. Introduction

Acute myeloid leukemia (AML) is a heterogeneous group of clonal aggressive hematologic malignancies characterized by differentiation block and increased proliferation of neoplastic cells of myeloid origins [1]. The presence of at least 20% myeloblasts in the bone marrow (BM) or peripheral blood (PB) is the main criterion of AML diagnosis as outlined in the 2016 World Health Organization (WHO) guidelines [1,2]. Exceptions are AML with specific cytogenetic abnormalities or nucleophosmin 1 (*NPM1*) mutated leukemias [3]. However, the 20% cut-off is arbitrary and is used for differential diagnosis with myelodysplastic syndromes (MDS), especially excess blasts 2 subtype [4]. MDS are a group of clonal premalignant hematological diseases characterized by ineffective hematopoiesis, progressive PB cytopenias, increased risk of developing AML, and poor overall survival (OS) [2]. MDS are heterogeneous in clinical presentation, cytogenetics and molecular signatures resulting in various outcomes with OS ranging from 5 years to 9 months [5]. Several genetic alterations frequently found in MDS can be present also in other hematological disorders and in healthy individuals because clonal hematopoiesis is commonly seen with aging [6,7]. Therefore, additional pathogenetic mechanisms are required for dysplastic hemopoiesis, and immune dysregulation can initiate or support dyspoiesis. Several studies have been investigating the prognostic impact of cytogenetic and molecular abnormalities in event-free survival (EFS) and OS of AML and MDS patients [8,9,10]. The European LeukemiaNet (ELN) has defined three risk categories in AML—favorable, intermediate, and adverse—based on the combination of specific genetic or chromosomal alterations, advanced age (>60 years old), or neutropenia [8]. Similarly, risk stratification of MDS patients is based on percentage of BM blasts, complete blood counts (CBC), and cytogenetic abnormalities [9,10].

Wilms’ tumor 1 (*WT1*), a zinc-finger transcription factor and an epigenetic modifier, has been proposed as prognostic marker of several solid and hematologic tumors because is frequently overexpressed in leukemias, lung, colon, or pancreatic cancers [11,12,13]. In physiological conditions, *WT1* is expressed at basal levels in CD34^+^CD38^−^ hematopoietic stem cells (HSCs) in the BM and is related to quiescence and stemness, while lineage-committed progenitors show undetectable *WT1* levels [11]. Conversely, *WT1* is frequently overexpressed in AML, MDS, and blast crisis of chronic myeloid leukemia, and expression levels are associated with increased blast counts, higher risk of progression and relapse, resistance to therapy, and poor OS [11,14,15,16,17].

In this study, we investigated the prognostic role of *WT1* expression levels and of a combined phenotypic and molecular score in AML and MDS patients for risk stratification.

## 2. Materials and Methods

### 2.1. Patients and Therapeutic Regimens

Whole PB or BM specimens were collected in ethylenediaminetetraacetic acid (EDTA) tubes for *WT1* expression level assessment or heparin tubes for immunophenotyping from patients after informed consent obtained in accordance with the Declaration of Helsinki [18]. A total of 71 patients were included in this retrospective study after received a diagnosis of AML or MDS, and chemotherapy as per international guidelines at the Hematology and Transplant Center, University Hospital ”San Giovanni di Dio e Ruggi d’Aragona” of Salerno, Italy. Risk stratification was calculated according to ELN or to the Revised International Prognostic Scoring System (IPSS-R) for AML or MDS, respectively [8,9,10]. International Working Group (IWG) consensus criteria were used to determine patients’ treatment response [19]. Clinical characteristics at baseline are summarized in Table 1 and Table 2.

Among AML patients (*n* = 46; mean age, 58 years old; range, 17–93; M/F, 26/20), 15 subjects received chemotherapy as per standard protocols with daunorubicin + cytarabine (Ara-C), FLANG (fludarabine + high-dose Ara-C + granulocyte colony-stimulating factor (G-CSF)), FLAG-IDA (FLAG plus idarubicin), or MEC (mitoxantrone, etoposide, and Ara-C); while 18 subjects were treated with hypomethylating agents, such as 5-azacitidine and decitabine, with or without venetoclax, a Bcl-2 inhibitor. Fifteen AML patients (eight in first CR, and seven in second CR) underwent to allogeneic hematopoietic stem cell transplantation (HSCT) after conditioning regimen with busulfan and melphalan. Among MDS patients (*n* = 25; mean age, 70 years old; range, 57–84; M/F, 17/8), five subjects were treated with lenalidomide because of del(5q), and 17 with hypomethylating agents with or without venetoclax.

### 2.2. WT1 Quantitative Assessment

*WT1* expression levels were quantified by real-time polymerase chain reaction (RT-PCR) at diagnosis, during treatment and follow-up. Mononuclear cells were freshly isolated from PB or BM samples by density gradient centrifugation using Lymphoprep (Axis-Shield Density Gradient Media, Oslo, Norway) and subsequently subjected to RNA extraction using QIAamp RNA Blood Mini Kit (Qiagen, Hilden, Germany) following manufacturer’s instructions. After RNA quantification using a BioSpectrometer (Eppendorf, Hamburg, Germany), at least 1 µg of RNA was used for cDNA reverse transcription (Ipsogen RT Kit Qiagen). Subsequently, WT1-mRNA quantitative assessment was performed using an ELN-certified Ipsogen *WT1* ProfilQuant Kit (Qiagen) following manufacturer’s instructions. 

### 2.3. Flow Cytometry

For immunophenotyping, 50 µL of fresh heparinized whole PB or BM was stained with antibodies according to the manufacturers’ instructions. The following antibodies were used for PB immunophenotyping: CD56, CD45, CD34, CD19, CD11b, CD3, CD8, CD71, CD33, CD16, SmIg-kappa, and SmIg-lambda. For BM immunophenotyping, the following antibodies were employed: CD3, CD7, CD5, CD19, CD34, CD16, CD11b, CD13, CD14, CD56, CD45, CD33, HLA-DR, CD117, SmIg-kappa, and SmIg-lambda. CD45dim blast phenotype was further studied for CD19, CD20, CD34, CD56, CD5, CD117, CD33, CD16, CD11b, CD36, CD13, HLA-DR, CD64, CD4, CD5, CD7, CD14, CD10, CD15, CD11a, CD11c, CD45RA, CD45RO, CD61, CD42b, TdT, and MPO expression. Manufacturer’s characteristics of used antibodies are summarized in Table 3. After 20 min incubation at room temperature, red cell lysis was performed with IO Test Lysing Solution (Beckman Coulter, Brea, CA, United States), cells were washed twice with phosphate-buffered saline (PBS) (IsoFlow Sheath Fluid, Beckman Coulter), and then resuspended in 500 µL PBS for acquisition.

Samples were acquired on a Navios or Navios/EX cytometer (Beckman Coulter), equipped with blue (488 nm), green (532 nm), and red (633 nm) lasers. Instrument daily quality control was carried out using Flow-Check Pro Fluorospheres (Beckman Coulter), and external quality control by UK NEQAS for Leucocyte Immunophenotyping. Compensation was monthly checked by a Beckman Coulter’s Specialist using Flow-Set and compensation kit (Beckman Coulter). Samples were run using the same PMT voltages, and at least 50,000–200,000 events were recorded. Post-acquisition analysis was carried out using Navios Software v1.3, Navios EX Software v2.0, or Kaluza Analysis Flow Cytometry Software v2.1.1 (Beckman Coulter).

Cell populations were first identified based on linear parameter (forward scatter area, FSC-A) and CD45 expression, and lymphocytes, monocytes, granulocytes, and immature cells were gated. Lymphocytes were further studied for T (CD3 or CD5 or CD7, CD4, and CD8), B (CD19, SmIg-kappa, and SmIg-lambda), and NK cell (CD56 and CD16) markers. CD33, CD14, CD11b, and CD56 expression was investigated on monocytes with additional CD13, CD36, CD64, CD15, and CD16 assessment in case of monocyte frequency was increased or showed aberrant marker expression. Maturation profiling of CD33^+^CD56^−^ granulocytes in the BM was carried using CD16 vs. CD11b expression. Normal CD34+ cells were gated for CD19, CD117, and CD33 for definition of lymphoid (CD19^+^) or myeloid (CD117^+^CD33^+^) progenitors. Hematogones were identified based on CD19, CD34, and CD45 expression (CD19^+^CD34^−^CD45^+/−^). CD45^dim^ blasts were further investigated with specific surface and intracellular markers which were also employed for monitoring minimal residual disease (MRD).

### 2.4. Statistical Analysis 

Data were analyzed using Prism (v.8.3.0; GraphPad software, La Jolla, CA, USA). For *WT1* quantification, *ABL* was employed as housekeeping gene for data normalization, and *WT1* levels were reported as *WT1* copy number/10^4^
*ABL* copies (normalized *WT1* expression). Normal expression levels were considered <50 or <250 copies in PB or BM, respectively, as previously reported [16,17,18,19,20]. Normalized blast count (NBC) was calculated as following: NBC = (%CD34^+^ cells + %immature cells + %blasts)/%granulocytes, using frequencies measured by flow cytometry. During monitoring, *WT1* was reported increased when levels were 2.5-fold higher than those documented at previous timepoint, while *WT1* was considered decreased when levels were 0.5-fold lower than those at previous timepoint. For flow cytometry data, populations were reported as percent of positive cells. Pearson analysis was employed for studying correlations between *WT1* levels and clinical and phenotypic features. Unpaired two-tailed *t*-tests for two group comparison and one-way analysis of variance (ANOVA) for three-group comparison were performed. Log-rank (Mantel-Cox) test was employed for survival analysis between groups. Chi-square test was employed for testing association between *WT1* expression levels and cytogenetic abnormalities. Multivariate linear regression was used for investigation of association of *WT1* levels with other clinical and biological features. A *p* < 0.05 was considered statistically significant.

## 3. Results

### 3.1. Correlation of WT1 Expression with Clinical, Phenotypic, and Molecular Features in AML

Whether to investigate associations between WT1-mRNA levels and AML disease severity and prognosis, *WT1* expression was assessed at diagnosis, during therapy, and follow-up, and levels were correlated with complete blood counts (CBCs) and flow cytometer counts (percentage of granulocytes, percentage of CD34^+^ cells, and NBC). In addition, percentage of BM blasts identified by light microscopy or flow cytometry was correlated with normalized *WT1* expression (Figure 1). No correlations with white blood cells (WBCs) were described (*r* = 0.1002; *p* = 0.2763), while a negative correlation was documented between normalized *WT1* levels and platelets (*r* = −0.2858; *p* = 0.0016) or hemoglobin levels (*r* = −0.2205; *p* = 0.0155) (Figure 1A). In contrast with findings described for WBC, a negative correlation between normalized *WT1* expression and granulocytes identified by flow cytometry was described (*r* = −0.3350; *p* = 0.0014). Moreover, *WT1* levels correlated with percentage of CD34^+^ cells (*r* = 0.3221; *p* = 0.0089) and NBC (*r* = 0.3383; *p* = 0.0011) (Figure 1B). Mean percentage of BM blasts by light microscopy at diagnosis was 43.3 ± 27.4% (range, 15–88%), while mean blast count by flow cytometry was 31.35 ± 30.8% (range, 0–94%). A positive correlation with normalized *WT1* levels was described for both light microscopy (*r* = 0.6032; *p* = 0.0063) and flow cytometry blast counts (*r* = 0.3578; *p* = 0.0348) (Figure 1C).

AML patients were then divided based on ELN risk category, and normalized *WT1* levels were compared among groups. Mean normalized *WT1* expression was 1718 ± 2551 copies (range, 3–7373 copies) in seven patients (15%) with ELN favorable risk, 3375 ± 7258 copies (range, 2–31,684 copies) in 25 subjects (54%) with intermediate risk, and 5960 ± 10,295 copies (range, 3–34,537 copies) in 13 (28%) ELN adverse risk patients. Despite patients with intermediate and adverse risk tended to have higher *WT1* levels compared to subjects with favorable risk, no statistically significant differences were described (*p* = 0.4662).

Patients were then divided based on the presence of somatic mutations in *FLT3* (fms like tyrosine kinase 3) or *NPM1*, and normalized *WT1* expression was compared among groups. Higher *WT1* levels were found in patients carrying somatic mutations in *FLT3* internal tandem duplication (ITD) (mean ± SD, 11,240 ± 13,127 copies; range, 2–31,684 copies) compared to those with *FLT3* wild type (mean ± SD, 2801 ± 6205 copies; range, 2–34,537 copies) (*p* = 0.0127). No differences were described between patients carrying somatic mutations in *NPM1* (mean ± SD, 2982 ± 5445 copies; range, 3–6100 copies) and those with *NPM1* wild type (mean ± SD, 4280 + 8651 copies; range, 2–34,537 copies) (*p* = 0.6728).

### 3.2. Prognostic Impact of WT1 Expression and Combined Score in AML 

In order to investigate prognostic impact of normalized *WT1* levels in AML, patients were first divided in two groups based on *WT1* expression at diagnosis ≥ cut-offs or within normal ranges (Figure 2A). Patients with increased *WT1* levels at diagnosis displayed a shorter OS compared to those with *WT1* levels < cut-offs (11.7 vs. 92.4 months, *WT1* ≥ cut-offs vs. *WT1* < cut-offs, respectively; *p* = 0.0002; hazard ratio (HR), 4.305; 95% confidential interval (CI), 1.983 to 9.344). Patients were also divided based on NBC > 0.5 and OS was compared between groups (Figure 2B); however, no statistically significant variations were described between patients with NBC > 0.5 at diagnosis and those with NBC ≤ 0.5 (10.23 vs. 38.23 months, NBC > 0.5 vs. NBC ≤ 0.5, respectively; *p* = 0.2396; HR, 0.6442; 95%CI, 0.2754 to 1.507). We then combined *WT1* expression levels and NBC at diagnosis, patients were divided in four groups, and OS was compared (Figure 2C). Patients with increased *WT1* levels and NBC had the shortest OS compared with those with only elevated *WT1* expression or NBC or compared with subjects with normal *WT1* levels and NBC ≤ 0.5 (4.43 vs. 17.97 vs. 123.17 vs. 78.1 months, *WT1* ≥ cut-off + NBC > 0.5 vs. *WT1* ≥ cut-off + NBC ≤ 0.5 vs. *WT1* < cut-off + NBC > 0.5 vs. *WT1* < cut-off + NBC ≤ 0.5, respectively; *p* < 0.0001).

Next, a prognostic score was developed by assigning a value of 1 if mutant *FLT3* ITD was present, *WT1* expression levels was ≥ cut-offs, and/or NBC > 0.5 (Figure 2D). Patients were divided in four groups according to this simple combined phenotypic and molecular score ranging from 0 to 3, and OS was compared between risk categories. Patients with a score of 3 displayed the shortest OS compared to other groups (78.1 vs. 27.3 vs. 14.3 vs. 1.3 months, score 0 vs. score 1 vs. score 2 vs. score 3, respectively; *p* = 0.0002). In multivariate analysis, *WT1* expression levels were significantly associated with the score (*p* = 0.0162), but not with NBC, *FLT3* or *NPM1* mutational status, cytogenetic abnormalities, sex, age, ELN risk stratification, or outcome.

### 3.3. Correlation of WT1 Expression with Clinical, Phenotypic, and Molecular Features in MDS Patients

Expression levels of *WT1* at diagnosis, during treatment, and follow-up were correlated with CBCs, flow cytometric counts, and percentage of blasts in our MDS patients (Figure 3). Similar to that reported in the AML cohort, no correlations with WBCs were described (*r* = −0.094; *p* = 0.3634), while negative correlations between normalized *WT1* levels and platelets (*r* = −0.2044; *p* = 0.0470) and hemoglobin levels (*r* = −0.3386; *p* = 0.0008) were confirmed in MDS patients (Figure 3A). Negative correlations between normalized *WT1* expression and granulocytes identified by flow cytometry were also described in MDS subjects (*r* = −0.3664; *p* = 0.0506) as documented in AML patients. In addition, *WT1* levels correlated with percentage of CD34^+^ cells (*r* = 0.8383; *p* < 0.0001) and NBC (*r* = 0.3700; *p* = 0.0482) (Figure 3B). Mean percentage of BM blasts by light microscopy at diagnosis was 6.6 ± 4.7% (range, 1–16%), and mean percentage of blasts by flow cytometry was 5.5 ± 7.8% (range, 0–38%). A positive correlation with normalized *WT1* levels was described for flow cytometry blasts (*r* = 0.7019; *p* < 0.0001), while no correlations were documented between *WT1* levels and percentage of blasts at diagnosis by light microscopy (*r* = 0.0428; *p* = 0.8579) (Figure 3C).

Higher *WT1* expression levels were documented in intermediate-2/high-risk MDS patients (mean ± SD, 3107 ± 4397 copies; range, 8–16,364 copies) compared to low-/intermediate-1 risk subjects (mean ± SD, 398.3 ± 770.6 copies; range, 0.4–2119 copies); however, no statistically significant variations were registered (*p* = 0.1266) likely because of the small number of patients in our MDS cohort.

### 3.4. Associations with Chromosomal Abnormalities

Associations between *WT1* levels and cytogenetic abnormalities were investigated in both AML and MDS patients (Figure 4). In AML cohort, 39 subjects were evaluable for the presence of chromosomal abnormalities, and 12 of them (31%) had *WT1* levels < cut-off at diagnosis, while remaining 27 subjects (69%) had increased levels (Figure 4A). Seven out of 12 AML patients with normal *WT1* expression had normal karyotype (58%), and five one or more chromosomal abnormalities (42%). Similarly, 16 out 27 AML subjects with increased *WT1* levels (59%) had normal karyotype, and 11 one or more chromosomal abnormalities (41%). In our MDS cohort, 19 subjects were evaluable for cytogenetic abnormalities (Figure 4B). Two out six patients with normal *WT1* levels did not show chromosomal abnormalities (33%), and four had one or more abnormalities (67%). Similarly, four out of 13 MDS patients with increased *WT1* levels had one chromosomal abnormality (31%), and nine one or more (69%). Therefore, no statistically significant differences were described in both AML and MDS cohorts by Chi-square test (*p* > 0.9999).

### 3.5. Prognostic Impact of WT1 Expression in MDS

Whether to investigate prognostic impact of normalized *WT1* levels in MDS, patients were divided in two groups based on *WT1* expression at diagnosis (Figure 5A). No statistically significant differences in OS were described between patients with increased *WT1* levels at diagnosis and those with *WT1* levels < cut-offs (25.9 vs. 20.7 months, *WT1* ≥ cut-offs vs. *WT1* < cut-offs, respectively; *p* = 0.0.2837; HR, 1.815; 95% CI, 0.6454 to 5.105). Patients were also divided based on NBC > 0.5 and OS was compared between groups (Figure 5B). A slight shorter OS was documented in patients with NBC > 0.5 at diagnosis compared to those with NBC ≤ 0.5 (9 vs. 35.7 months, NBC > 0.5 vs. NBC ≤ 0.5, respectively; *p* = 0.0613; HR, 0.3827; 95% CI, 0.0964 to 1.518); however, statistical significance was not reached likely because of the small number of patients in each group (NBC ≤ 0.5, *n* = 17; NBC > 0.5, *n* = 5). We then divided patients in four groups based on the combination of *WT1* expression levels and NBC at diagnosis, and OS was compared among groups (Figure 5C). Patients with increased *WT1* levels and NBC had the shortest OS compared with those with elevated *WT1* expression and NBC ≤ 0.5 or those subjects with normal *WT1* levels and NBC ≤ 0.5 (9 vs. 36.7 vs. 20.7 months, *WT1* ≥ cut-off + NBC > 0.5 vs. *WT1* ≥ cut-off + NBC ≤ 0.5 vs. *WT1* < cut-off + NBC ≤ 0.5, respectively; *p* = 0.1648). Statistical significance was not reached likely because of the small number of patients in each group (WT1 ≥ cut-off + NBC > 0.5, *n* = 5; *WT1* ≥ cut-off + NBC ≤ 0.5, *n* = 8; *WT1* < cut-off + NBC ≤ 0.5, *n* = 9). In multivariate analysis, no statistically significant associations were described.

## 4. Discussion

WT1 is expressed in normal hemopoiesis at basal levels while is overexpressed in the majority of AML [11]. Indeed, *WT1* measurement by PCR has been proposed for monitoring minimal residual diseases in AML patients after allogeneic HSCT because *WT1* levels strongly correlate with chimerism and disease relapse [21,22,23,24]. In addition, *WT1* expression levels might also be used as prognostic factor in MDS [16,17]. In this retrospective real-world evidence study, we have investigated prognostic value of *WT1* expression in AML and MDS, alone and in combination with risk categories, CBCs, flow cytometry counts, and molecular biology features.

In normal hemopoiesis, *WT1* is expressed at basal levels in quiescent cells and is correlated with stemness as differentiated and mature cells express *WT1* at very low levels [11,25,26]. Conversely, leukemic cells have increased *WT1* levels, and monitoring of minimal residual diseases using *WT1* quantification is a specific and sensitive biomarker of disease relapse or progression, especially after allogeneic HSCT [11,23,27,28]. In our study, we confirmed that *WT1* inversely correlated with normal hemopoiesis as a negative correlation with platelet count and hemoglobin levels was described in both AML and MDS patients. For WBC, we showed that flow cytometry is more sensitive and specific to distinguish normal hemopoiesis from neoplastic counterpart, as WBC counted using an automated hemocytometer did not correlate with *WT1* levels. By contrast, percentage of granulocytes identified by flow cytometry using linear parameters and CD45 expression negatively correlated with *WT1* levels, while positively associated with CD34^+^, immature, and CD45^dim^ cells which represent the neoplastic counterpart. BM blast count by light microscopy was also correlated with *WT1* levels in AML, while not in MDS. Conversely, blast count by flow cytometry was positively associated with *WT1* levels also in MDS patients, confirming flow cytometry as a more sensitive and specific tool for identification of leukemic cells compared to light microscopy that is operator dependent [29,30,31].

*WT1* has been largely reported to be overexpressed in all types of leukemias both of myeloid and lymphoid origins, as well as during blast crisis of chronic myeloid leukemia, and increased *WT1* levels are associated with poor outcomes because of higher incidence of relapse and resistance to standard chemotherapy [11,14,16,17]. Indeed, a 2-log reduction of *WT1* levels after induction and consolidation therapy is related to a better outcome and a reduced relapse rate in AML patients [15]. Moreover, *WT1* levels <50 copies/10^4^
*ABL* copies in PB or <250 copies in BM are associated with complete remission (CR), while *WT1* levels above the cut-offs are an early predictor of disease progression [15,20]. However, the exact role of *WT1* in leukemogenesis and disease progression is still unclear. In addition, *WT1* has been reported to be increased in MDS, especially in those patients with elevated blast counts and higher risk of AML development [11,27]. A limited number of studies proposes *WT1* as a prognostic factor in MDS, as peripheral blood WT1-mRNA levels correlate with IPSS-R risk category and outcomes [16,17]. According to published data, we confirmed the prognostic role of *WT1* expression in both AML and MDS patients, regardless type of disease and risk category. However, OS did not statistically differ between MDS patients with increased *WT1* levels and those with normal levels, likely because of the small number of subjects included in each group.

Diagnosis and prognosis of MDS patients are still challenging and cannot rely on a single clinical, phenotypic, or molecular feature; indeed, prognostic scores are calculated based on cytogenetic abnormalities, BM blasts, and CBCs [8,9,32]. However, we showed that CBCs did not correlate with *WT1*-mRNA levels in our cohort, while flow cytometry counts were highly associated. Therefore, a better prognostic definition could be achieved by combining clinical, molecular, and flow cytometry features. Previous studies have shown the utility of flow cytometry scoring system for differential diagnosis and prognosis of MDS [29]. Here, we added evidence for inclusion of flow cytometry counts in diagnostic and prognostic definition of AML and MDS patients. Combination of *WT1* expression levels, NBC, and *FLT3* mutational status allowed a better risk stratification in AML patients. Similarly, risk stratification based only on *WT1* levels and NBC might identify a subgroup of MDS subjects with a poorer prognosis. *WT1* mutations are frequently found in de novo AML, especially in younger patients and in *FLT3*-ITD or *CEBPA* mutated leukemias [11,17,26,33,34]. Recently, mutant *WT1* has been also associated with *NPM1* as secondary mutations; while when *WT1* occurs as a secondary mutation, the most common dominant alterations are in *DNMT3A*, which also negatively affects prognosis of AML patients, *PHF6*, *FLT3*, and *CEBPA* [26]. However, these comprehensive studies on genomic landscape of *WT1* mutant AML do not shade lights on the role of *WT1* in leukemogenesis, and it remains unclear if mutations in *WT1* and associated genes are drivers or passengers. In our study, we focused on *WT1* expression rather than mutational status. In addition, we correlated WT1-mRNA levels with the presence of cytogenetic abnormalities showing that there was no association between *WT1* expression levels and cytogenetic alterations. These preliminary findings suggested that *WT1* overexpression might not have effects on genomic instability favoring chromosomal alterations.

## 5. Conclusions

WT1-mRNA levels have been proposed as a diagnostic and prognostic marker of AML and MDS [11,16]. Indeed, *WT1* levels can identify MDS with higher risk of AML development, relapsed/refractory AML, and early relapse after allogeneic stem cell transplantation [24]. WT1-mRNA expression is more sensitive and specific as hematological biomarker compared to *WT1* mutational status [26]. Moreover, as we are improving our understanding of disease biology, prognostic definition of AML and MDS should consider clinical, phenotypic, and molecular features [8,9]. Here, we proposed a simple score based on *WT1* expression levels, NBC by flow cytometry, and *FLT3* mutational status for risk stratification of AML patients, while *WT1* expression levels and NBC might identify a subgroup of MDS patients with poorer prognosis. However, our findings need further validation in larger cohorts and prospective studies.

## Figures and Tables

**Figure 1 biomedicines-09-00387-f001:**
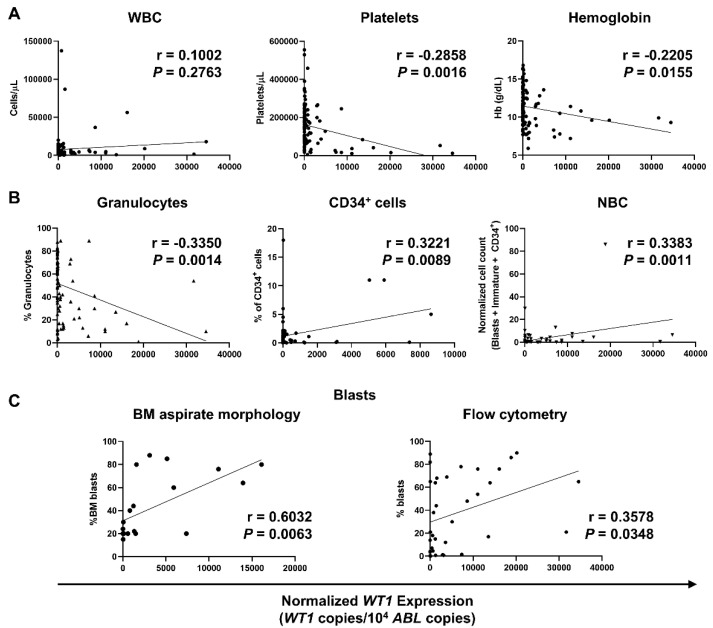
Correlations of normalized *WT1* levels with complete blood counts (CBCs), flow cytometry parameters, and percentage of blasts in AML patients. *WT1* expression was normalized using *ABL* as housekeeping gene, and levels were reported as *WT1* copies/10^4^
*ABL* copies. (**A**) Pearson correlation analysis between normalized *WT1* expression and CBCs, such as white blood cells (WBC), platelets, and hemoglobin (Hb) levels. (**B**) Correlations with percentage of granulocytes, CD34+ cells, and normalized blast count (NBC) by flow cytometry. (**C**) Correlations with bone marrow (BM) blasts identified by light microscopy (left) or by flow cytometry (right).

**Figure 2 biomedicines-09-00387-f002:**
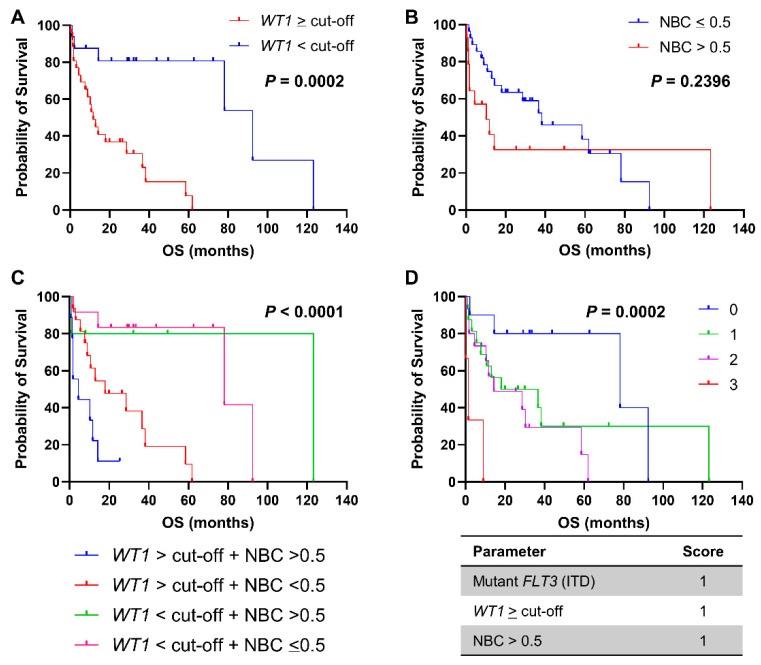
Clinical outcomes of AML patients. Overall survival (OS) of AML patients is reported based on (**A**) normalized *WT1* expression ≥ cut-off (50 copies in peripheral blood or 250 copies in bone marrow samples), (**B**) normalized blast count (NBC) > 0.5, or (**C**) combination of these two features. (**D**) A prognostic score was calculated based on the presence of mutant FLT3, *WT1* ≥ cut-off, and NBC > 0.5, and OS were compared among groups. A *p* < 0.05 was considered statistically significant.

**Figure 3 biomedicines-09-00387-f003:**
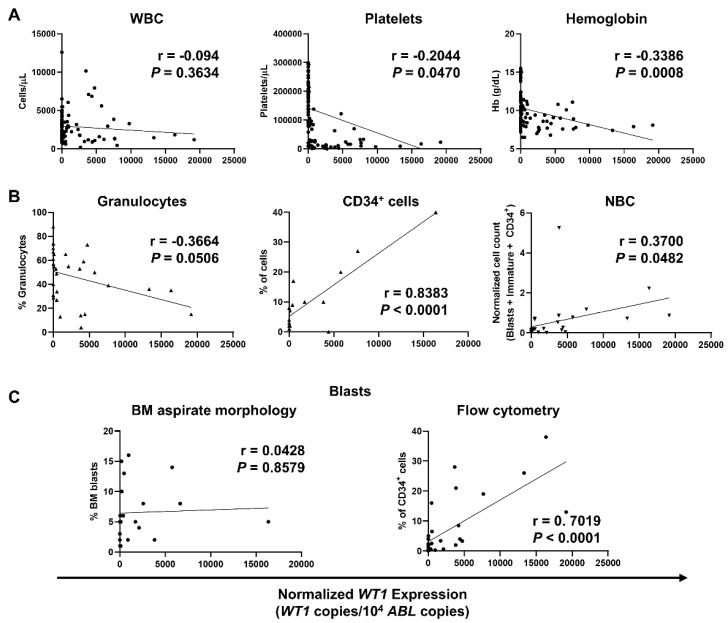
Correlations of normalized *WT1* levels with complete blood counts (CBCs), flow cytometry parameters, and percentage of blasts in MDS patients. *WT1* expression was normalized using *ABL* as housekeeping gene, and levels were reported as *WT1* copies/10^4^
*ABL* copies. (**A**) Pearson correlation analysis between normalized *WT1* expression and CBCs, such as white blood cells (WBC), platelets, and hemoglobin (Hb) levels. (**B**) Correlations with percentage of granulocytes, CD34+ cells, and normalized blast count (NBC) by flow cytometry. (**C**) Correlations with bone marrow (BM) blasts identified by light microscopy (left) or by flow cytometry (right).

**Figure 4 biomedicines-09-00387-f004:**
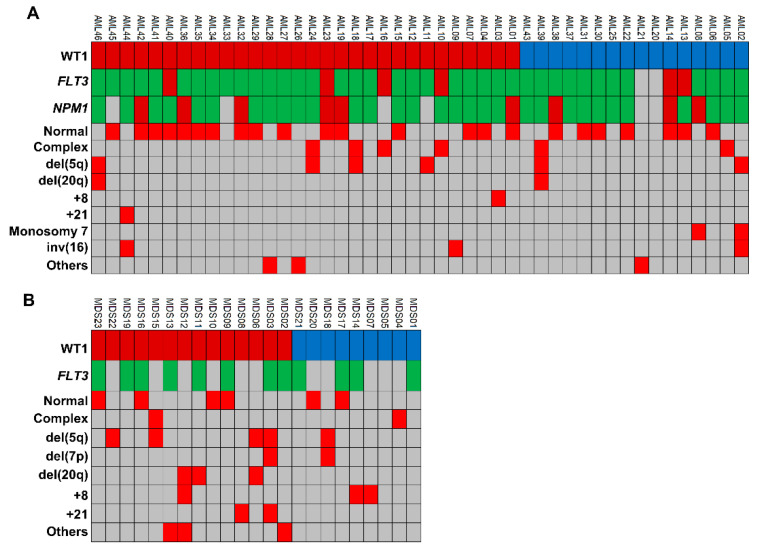
Chromosomal abnormalities in our acute myeloid leukemia (AML) and myelodisplastic syndrome (MDS) patients. (**A**) AML and (**B**) MDS patients (UPN) were screened for the presence of cytogenetic abnormalities and divided based on *WT1* expression levels at diagnosis (WT1 peripheral blood cut-off, 50 copies; and *WT1* bone marrow cut-off, 250 copies). Normal expression *WT1* levels are reported in blue, while increased in dark red. For each patient, cytogenetic alteration detected by karyotype or fluorescence in situ hybridization analysis are reported in red. Screening for *FLT3* and *NPM1* mutations was also performed and reported in green for wild type genes, or in red for mutant forms. Light grey is used when marker was not tested or not present.

**Figure 5 biomedicines-09-00387-f005:**
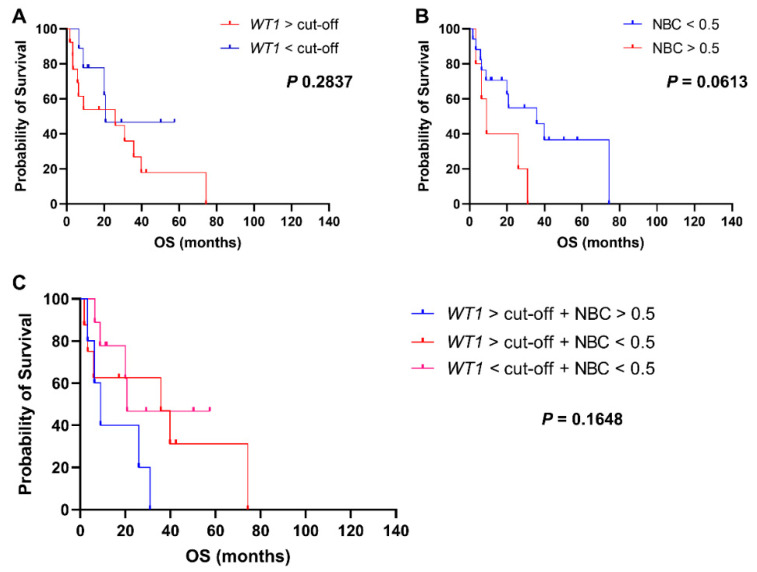
Clinical outcomes of MDS patients. Overall survival (OS) of MDS patients is reported based on (**A**) normalized *WT1* expression ≥ cut-off (50 copies in peripheral blood or 250 copies in bone marrow samples), (**B**) normalized blast count (NBC) > 0.5, or (**C**) combination of these two features.

**Table 1 biomedicines-09-00387-t001:** AML patients’ characteristics at baseline.

Characteristics	*n* = 46	(Range)
Age, years	58	(17–93)
M/F	26/20	
FAB classification		
M0–M1	25	54%
M2	2	4%
M4–M5	9	20%
Secondary/others	10	22%
ELN risk stratification		
Favorable	7	15%
Intermediate	25	54%
Adverse	13	28%
Not evaluable	1	3%
Cytogenetic abnormalities		
Isolated del(5q)	1	2%
Isolated del(1q)	1	2%
Isolated monosomy 7	1	2%
Any isolated trisomy	2	4%
Isolated inv (16)	1	2%
≥2 chromosomal abnormalities	10	22%
Normal karyotype	23	50%
Not evaluable	5	11%
Not performed	2	4%
Molecular features		
*FLT3* (mutated/WT)	6/38	
*NPM1* (mutated/WT)	9/33	
*FLT3* ^+^ *NPM1* ^+^	2	
*WT1* (copies/10^4^ *ABL* copies)	3780.3	(2–34,537)
BM blasts, %	43.8	(6–90)
WBC (cells/μL)	27,867.30	(510–169,000)
NBC	4.38	(0.001–86)
Follow-up, months	27.2	(0.33–123.2)
Dead/Alive	29/16	
First-line therapy	43/46	
Hypomethylating agents ± venetoclax	18	42%
Daunorubicin + Ara-C	15	35%
Others	10	23%

Abbreviations: AML, acute myeloid leukemia; FAB, French-American-British; ELN, European LeukemiaNet; *FLT3*, fms related tyrosine kinase 3; *NPM1*, nucleophosmin 1; *WT1*, Wilms’ tumor 1; BM, bone marrow; WBC, white blood cells; NBC, normalized blast count; Ara-C, cytarabine.

**Table 2 biomedicines-09-00387-t002:** MDS patients’ characteristics at baseline

Characteristics	*n* = 25	(Range)
Age, years	70	(57–84)
M/F	17/8	
WHO classification		
MLD	7	28%
EB-1	10	40%
EB-2	4	16%
CMML	4	16%
IPSS risk stratification		
Low	3	4%
Intermediate-1	6	20%
Intermediate-2	15	60%
High	1	4%
Cytogenetic abnormalities		
Isolated del(5q)	1	4%
Isolated del(20q)	1	4%
Isolated der(13)	1	4%
Any isolated trisomy	4	16%
≥2 chromosomal abnormalities	6	24%
Normal karyotype	6	24%
Not evaluable	3	12%
Not performed	3	12%
*WT1* (copies/10^4^ *ABL* copies)	2013	(0.4–16,364)
BM blasts, %	6.6	(1–16)
WBC (cells/μL)	4207	(1180–12,580)
Hb (g/dL)	9.4	(7.2–13.9)
Platelets (/μL)	86,850	(6000–256,000)
NBC	0.6	(0.01–5.25)
Follow-up, months	25.1	(1.7–74.3)
Dead/Alive	15/7	
First-line therapy		
Hypomethylating agents ± venetoclax	21	84%
Supportive therapies	4	16%

Abbreviations: MDS, myelodysplastic syndromes; WHO, World Health Organization; MLD, multilineage dysplasia; EB, excess blasts; CMML, chronic myelo-monocytic leukemia; IPSS, international prognostic scoring system; *WT1*, Wilms’ tumor 1; BM, bone marrow; WBC, white blood cells; Hb, hemoglobin; NBC, normalized blast count.

**Table 3 biomedicines-09-00387-t003:** Antibodies for immunophenotyping

Antigen	Fluorochrome	Clone
CD3	APC	UCHT1
CD4	PE	13B8.2
CD8	APC-A750	B9.11
CD5	PC7	BL1a
CD7	PC7	8H8.1
CD19	PC5.5	J3-119
CD20	PB	B9E9
SmIg-kappa/SmIg-lambda/CD19	FITC/PE/ECD	Polyclonal/Polyclonal/J3-119
CD56	ECD	N901
CD16	PB	3G8
CD10	PC7	ALB1
CD11a	FITC	25.3
CD11b	PC7	Bear1
CD11c	PE	BU15
CD13	PC5.5	Immu103.44
CD14	APC-A750	RMO52
CD15	PE	80H5
CD33	APC	D3HL60.251
CD34	APC700	581
CD36	FITC	FA6.152
CD64	ECD	22
CD117	PE	104D2D1
HLA-DR	FITC	Immu-357
CD45	KO	J33
CD45RA	FITC	ALB11
CD45R0	ECD	UCHL1
CD71	FITC	YDJ1.2.2
CD61	FITC	SZ21
CD42b	PE	SZ2
TdT	FITC	HT1+HT4+HT8+HT9
MPO	PE	CLB-MPO-1

Abbreviations: CD, cluster of differentiation; TdT, terminal deoxynucleotidyl transferase; MPO, myeloperoxidase; APC, allophycocyanin; PE, phycoerythrin; PC, phycoerythrin cyanin; PB, pacific blue; FITC, fluorescein isothiocyanate; ECD, PE-Texas Red; KO, krome orange.

## Data Availability

Data are available upon request by the authors.

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
