# Peer review of "WT1 Expression Levels Combined with Flow Cytometry Blast Counts for Risk Stratification of Acute Myeloid Leukemia and Myelodysplastic Syndromes"

_biomedicines, 2021, doi:10.3390/biomedicines9040387_

Round 1

Reviewer 1 Report

Giudice et al. present data on the influence of WT1 on outcomes in AML and MDS. Whereas the concept is interesting, there are areas that need significant improvement. The authors should consider the following.

1. There are 46 patients with acute myeloid leukemia. The authors have provided us risk stratification based on ELN 2017.  Based on table #1, it seems that a total of 15 patients were tested for FLT3 mutation. The stratification per ELN 2017 is provided for 41 patients.  I am not sure how to interpret the stratification based on the fact that FLT3 was only tested in 15 patients.  Furthermore, only 25 patients in the AML cohort were evaluable for cytogenetic abnormalities.  Again it is difficult to interpret the ELN 2017 risk ratification without having cytogenetic abnormalities available for all patients.  Without detailed cytogenetic and molecular data available for all or majority of the patients, it is very challenging to interpret the significance of WT1 in these patients.
2.  The authors provide a scoring system for AML which comprises of FLT3 mutation, WT1 level as well as NBC.  There are 2 issues with the FLT3 competent of this scoring system.  The main issue is that there were only 3 patients with the mutation. This number is very small to draw any significant conclusions.  Secondly, it includes both ITD as well as TKD.  The significance of TKD is unclear but is thought to be less prognostic than ITD. The authors are being ambitious in using that small of a number with both ITD and TKD to develop a scoring system.
3. The authors should provide a rationale as to why they just looked at NPM1 and FLT3 mutation status.  Other mutations are also independently associated with outcomes in AML and I am not sure why they were excluded from this analysis.  Those mutations would include bi-allelic CEBPA, TP53, ASXL1 and RUNX1.
4. When the authors provide associations with chromosomal abnormalities for AML, it is challenging to draw any clear associations.  It seems like in patients with low levels of WT1, 43% patients have multiple chromosomal abnormalities and in patients with increased levels of WT1, 61% have normal karyotype.  Again the numbers are too small to draw any solid conclusions. However, one would expect higher number of chromosomal abnormalities with increased WT1 levels based on what the authors are proposing.
5. For the MDS cohort, there is no significant difference in results based on WT1 level.  It is unclear why the authors are saying that a scoring system incorporating WT1 level, FLT3 mutation status and NBC is a better prognostic tool for MDS.  The role of FLT3 in MDS is unclear and the results of the AML cohort should not be applied to MDS in this setting.
6. The numbers are small but it would be helpful to do a multivariable analysis to really understand whether WT1 is independently associated with outcomes in both AML and MDS.

Author Response

Giudice et al. present data on the influence of WT1 on outcomes in AML and MDS. Whereas the concept is interesting, there are areas that need significant improvement. The authors should consider the following.

Comment 1: There are 46 patients with acute myeloid leukemia. The authors have provided us risk stratification based on ELN 2017.  Based on table #1, it seems that a total of 15 patients were tested for FLT3 mutation. The stratification per ELN 2017 is provided for 41 patients.  I am not sure how to interpret the stratification based on the fact that FLT3 was only tested in 15 patients.  Furthermore, only 25 patients in the AML cohort were evaluable for cytogenetic abnormalities.  Again it is difficult to interpret the ELN 2017 risk ratification without having cytogenetic abnormalities available for all patients.  Without detailed cytogenetic and molecular data available for all or majority of the patients, it is very challenging to interpret the significance of WT1 in these patients.

Response to Comment 1: We thank the Reviewer for the comment, and we apologize for missing data in the previous version of this manuscript due to the fact that we have used information on digital databases. Therefore, we carefully went through printed clinical data for searching missing information and we have integrated them accordingly.

Based on updated data on Table 1, a total of 44/46 AML patients were tested for FLT3 mutations, 42/46 for NPM1 mutational status, and 44/46 for cytogenetic abnormalities. Therefore, ELN risk stratification was possible for 45/46 AML patients in our cohort.

Comment 2: The authors provide a scoring system for AML which comprises of FLT3 mutation, WT1 level as well as NBC.  There are 2 issues with the FLT3 competent of this scoring system.  The main issue is that there were only 3 patients with the mutation. This number is very small to draw any significant conclusions.  Secondly, it includes both ITD as well as TKD.  The significance of TKD is unclear but is thought to be less prognostic than ITD. The authors are being ambitious in using that small of a number with both ITD and TKD to develop a scoring system.

Response to Comment 2: We thank the Reviewer for these comments. We have increased the number of FLT3 mutant patients and statistical significance was maintained. As suggested, we have removed FLT3 TKD from the score. On page 7, lines 17-24, sentences were updated as following “Patients were then divided based on the presence of somatic mutations in FLT3 (fms like tyrosine kinase 3) or NPM1, and normalized WT1 expression was compared among groups. Higher WT1 levels were found in patients carrying somatic mutations in FLT3 internal tandem duplication (ITD) (mean+SD, 11,240+13,127 copies; range, 2-31,684 copies) compared to those with FLT3 wild type (mean+SD, 2,801+6,205 copies; range, 2-34,537 copies) (P = 0.0127). No differences were described between patients carrying somatic mutations in NPM1 (mean+SD, 2,982+5,445 copies; range, 3-16,100 copies) and those with NPM1 wild type (mean+SD, 4,280+8,651 copies; range, 2-34,537 copies) (P = 0.6728).”

We have changed Figure 2D accordingly.

Comment 3: The authors should provide a rationale as to why they just looked at NPM1 and FLT3 mutation status.  Other mutations are also independently associated with outcomes in AML and I am not sure why they were excluded from this analysis.  Those mutations would include bi-allelic CEBPA, TP53, ASXL1 and RUNX1.

Response to Comment 3: We agree with the Reviewer’s point, and, at the time of writing, we had those data only for a small subgroup of patients because NGS was not routinely performed before 2018. Therefore, we did not exclude data from analysis, we just used available information at the time of diagnosis.

Comment 4: When the authors provide associations with chromosomal abnormalities for AML, it is challenging to draw any clear associations.  It seems like in patients with low levels of WT1, 43% patients have multiple chromosomal abnormalities and in patients with increased levels of WT1, 61% have normal karyotype. Again the numbers are too small to draw any solid conclusions. However, one would expect higher number of chromosomal abnormalities with increased WT1 levels based on what the authors are proposing.

Response to Comment 4: We thank the Reviewer for the comment, and we have changed paragraph 3.4 and Figure 4 according with new data and results. On page 10, lines 7-19, the following text was added “Associations between WT1 levels and cytogenetic abnormalities were investigated in both AML and MDS patients (Figure 4). In AML cohort, 39 subjects were evaluable for the presence of chromosomal abnormalities, and 12 of them (31%) had WT1 levels < cut-off at diagnosis, while remaining 27 subjects (69%) had increased levels (Figure 4A). Seven out of 12 AML patients with normal WT1 expression had normal karyotype (58%), and five one or more chromosomal abnormalities (42%). Similarly, 16 out 27 AML subjects with increased WT1 levels (59%) had normal karyotype, and 11 one or more chromosomal abnormalities (41%). In our MDS cohort, 19 subjects were evaluable for cytogenetic abnormalities (Figure 4B). Two out six patients with normal WT1 levels did not show chromosomal abnormalities (33%), and four had one or more abnormalities (67%). Similarly, four out of 13 MDS patients with increased WT1 levels had one chromosomal abnormality (31%), and nine one or more (69%). Therefore, no statistically significant differences were described in both AML and MDS cohorts by Chi-square test (P > 0.9999).”

Comment 5: For the MDS cohort, there is no significant difference in results based on WT1 level.  It is unclear why the authors are saying that a scoring system incorporating WT1 level, FLT3 mutation status and NBC is a better prognostic tool for MDS.  The role of FLT3 in MDS is unclear and the results of the AML cohort should not be applied to MDS in this setting.

Response to Comment 5: We agree with the Reviewer, and we have removed that sentence in our discussion.

Comment 6: The numbers are small but it would be helpful to do a multivariable analysis to really understand whether WT1 is independently associated with outcomes in both AML and MDS.

Response to Comment 6: We thank the Reviewer for the comment. We have added multivariate analysis results accordingly: on page 9, lines 1-4 “In multivariate analysis, WT1 expression levels were significantly associated with the score (P = 0.0162), but not with NBC, FLT3 or NPM1 mutational status, cytogenetic abnormalities, sex, age, ELN risk stratification, or outcome.”

On page 11, line 28, the following text was added “In multivariate analysis, no statistically significant associations were described.”

Reviewer 2 Report

Giudice et al. present a study evaluating the role of WT1 expression in acute myeloid leukemia (AML) and myelodysplastic syndromes (MDS) risk stratification. The authors quantify WT1 levels in a cohort of 46 AML and 25 MDS and correlate these data with clinical, phenotypic and molecular features, providing an integrated approach for a better prognostic definition in these diseases.

The issue is well introduced and the methodological approach is properly described. Nevertheless, certain aspects are not clear and some conclusions are not supported by the results observed:

A) Page 2, lines 25-27. No data about the AML progression of MDS patients are reported in the present study.

B) Page 2, line 30 and page 3, line 7. It is not clear the sample type used in the study: only PB (page 2), PB or BM (page 3)?

C) Page 3, Table 1 and page 10, Figure 4A. It is not clear the number of AML patients carrying a FLT3 mutational analysis: 15 (Table 1) or 17 (Figure 4A)?

D) Page 12, lines 43-45. The authors should highlight that only a subgroup of their AML cases presents a FLT3 molecular evaluation, so their observation does not refer to the entire AML cohort.

E) Page 12, lines 46-47. The authors, on the basis of the findings in the AML cohort, propose the FLT3 mutational status as an additional prognostic tool in MDS patients. The proposal is not supported by experimental data in these diseases.

F) Page 13, lines 1-4. The association between WT1-mRNA levels and the presence of cytogenetic abnormalities is not supported by a statistical analysis.

G) Page 13, lines 4-6. As reported in Figure 4, WT>cut-off cases include only 5 AML cases carrying mutational alterations (FLT3 and/or NPM1), 7 AML and MDS cases showing cytogenetic alterations. No statistically significant differences are shown about the occurrence of these events between WTcut-off cases. So, I think that this declaration is a bit much.

H) Page 13, lines 14-18. No data about the FLT3 mutational status in MDS cohort are shown in this paper, so the authors can only include FLT3 mutational evaluation in the score proposed for AML patients.

I) Page 13, lines 17-18. No data about minimal residual disease monitoring are reported in this study. The assertion is inappropriate.

Author Response

Giudice et al. present a study evaluating the role of WT1 expression in acute myeloid leukemia (AML) and myelodysplastic syndromes (MDS) risk stratification. The authors quantify WT1 levels in a cohort of 46 AML and 25 MDS and correlate these data with clinical, phenotypic and molecular features, providing an integrated approach for a better prognostic definition in these diseases.

The issue is well introduced and the methodological approach is properly described. Nevertheless, certain aspects are not clear and some conclusions are not supported by the results observed:

Comment 1: Page 2, lines 25-27. No data about the AML progression of MDS patients are reported in the present study.

Response to Comment 1: We thank the Reviewer for this comment, and we have changed the sentence as following “In this study, we investigated the prognostic role of WT1 expression levels and of a combined phenotypic and molecular score in AML and MDS patients for risk stratification.”

Comment 2: Page 2, line 30 and page 3, line 7. It is not clear the sample type used in the study: only PB (page 2), PB or BM (page 3)?

Response to Comment 2: We apologize for missing information on page 2. We have used both PB and BM specimens. On page 2, lines 30-33, the following text was added “Whole PB or BM specimens were collected in ethylenediaminetetraacetic acid (EDTA) tubes for WT1 expression level assessment or heparin tubes for immunophenotyping from patients after informed consent obtained in accordance with the Declaration of Helsinki [18].”

Comment 3: Page 3, Table 1 and page 10, Figure 4A. It is not clear the number of AML patients carrying a FLT3 mutational analysis: 15 (Table 1) or 17 (Figure 4A)?

Response to Comment 3: We thank the Reviewer for the comment, and we have changed Figure 4 showing all patients in both cohorts.

Comment 4: Page 12, lines 43-45. The authors should highlight that only a subgroup of their AML cases presents a FLT3 molecular evaluation, so their observation does not refer to the entire AML cohort.

Response to Comment 4: We thank the Reviewer for the comment, and we apologize for missing data in the previous version of this manuscript due to the fact that we have used information on digital databases. Therefore, we carefully went through printed clinical data for searching missing information and we have integrated them accordingly.

Based on updated data on Table 1, a total of 44/46 AML patients were tested for FLT3 mutations, 42/46 for NPM1 mutational status, and 44/46 for cytogenetic abnormalities. Therefore, ELN risk stratification was possible for 45/46 AML patients in our cohort.

Comment 5: Page 12, lines 46-47. The authors, on the basis of the findings in the AML cohort, propose the FLT3 mutational status as an additional prognostic tool in MDS patients. The proposal is not supported by experimental data in these diseases.

Response to Comment 5: We agree with the Reviewer, and we have removed that proposal from our discussion.

Comment 6: Page 13, lines 1-4. The association between WT1-mRNA levels and the presence of cytogenetic abnormalities is not supported by a statistical analysis.

Response to Comment 6: We thank the Reviewer for the comment, and we have changed paragraph 3.4 and Figure 4 according with new data and results. On page 10, lines 7-19, the following text was added “Associations between WT1 levels and cytogenetic abnormalities were investigated in both AML and MDS patients (Figure 4). In AML cohort, 39 subjects were evaluable for the presence of chromosomal abnormalities, and 12 of them (31%) had WT1 levels < cut-off at diagnosis, while remaining 27 subjects (69%) had increased levels (Figure 4A). Seven out of 12 AML patients with normal WT1 expression had normal karyotype (58%), and five one or more chromosomal abnormalities (42%). Similarly, 16 out 27 AML subjects with increased WT1 levels (59%) had normal karyotype, and 11 one or more chromosomal abnormalities (41%). In our MDS cohort, 19 subjects were evaluable for cytogenetic abnormalities (Figure 4B). Two out six patients with normal WT1 levels did not show chromosomal abnormalities (33%), and four had one or more abnormalities (67%). Similarly, four out of 13 MDS patients with increased WT1 levels had one chromosomal abnormality (31%), and nine one or more (69%). Therefore, no statistically significant differences were described in both AML and MDS cohorts by Chi-square test (P > 0.9999).”

On page 13, lines 9-13, the text was changed as follows “In our study, we focused on WT1 expression rather than mutational status. In addition, we correlated WT1-mRNA levels with the presence of cytogenetic abnormalities showing that there was no association between WT1 expression levels and cytogenetic abnormalities. These preliminary findings suggested that WT1 overexpression might not have effects on genomic stability favoring chromosomal alterations.”

Comment 7: Page 13, lines 4-6. As reported in Figure 4, WT>cut-off cases include only 5 AML cases carrying mutational alterations (FLT3 and/or NPM1), 7 AML and MDS cases showing cytogenetic alterations. No statistically significant differences are shown about the occurrence of these events between WTcut-off cases. So, I think that this declaration is a bit much.

Response to Comment 7: Please refer to Response to Comment 6.

Comment 8: Page 13, lines 14-18. No data about the FLT3 mutational status in MDS cohort are shown in this paper, so the authors can only include FLT3 mutational evaluation in the score proposed for AML patients.

Response to Comment 8: We thank the Reviewer for the comment, and we have rephrased our conclusions as following “Here, we proposed a simple score based on WT1 expression levels, NBC by flow cy-tometry, and FLT3 mutational status for risk stratification of AML patients, while WT1 expression levels and NBC might identify a subgroup of MDS patients with poorer prognosis.”

Comment 9: Page 13, lines 17-18. No data about minimal residual disease monitoring are reported in this study. The assertion is inappropriate.

Response to Comment 9: Please refer to Response to Comment 9.

Round 2

Reviewer 1 Report

I appreciate the authors edits.